# Is there a relationship between forest fires and deforestation in the Brazilian Amazon?

**Cássio Furtado Lima**[1☉], **Fillipe Tamiozzo Pereira Torres**[2☉], **Luciano José Minette**[2☉], **Fernanda Araujo Lima**[2‡], **Roldão Carlos Andrade Lima**[3☉]*, **Michel Keisuke Sato**[4☉], **Arthur Araújo Silva**[2‡], **Bruno Leão Said Schettini**[2‡], **Francisco de Assis Costa Ferreira**[2‡], **Mateus Xavier Lima Machado**[2‡]

**1** Ananindeua Campus, Federal Institute of Education, Science and Technology of Pará (IFPA), Ananindeua, Pará, Brazil, **2** Department of Production and Mechanical Engineering/Department of Forestry Engineering, Federal University of Viçosa (UFV), Viçosa, Minas Gerais, Brazil, **3** Center of Agrarian Sciences, State University of the Tocantina Region of Maranhão (UEMASUL), Imperatriz, Maranhão, Brazil, **4** Bragança Campus, Federal Institute of Education, Science and Technology of Pará (IFPA), Bragança, Pará, Brazil

☉ These authors contributed equally to this work.
‡ FAL, AAS, BLSS, FACF and MXLM also contributed equally to this work.
* roldao.carlos@outlook.com

**Data Availability Statement:** All relevant data are within the manuscript and its Supporting Information files.

## Abstract

The Brazilian Legal Amazon is an extensive territory in which different factors influence the dynamics of forest fires. Currently, the Brazilian government has two tools in the public domain and free of charge, *PRODES* and *BDQueimadas*, to monitor and make decisions to combat deforestation and forest fires. This work aimed to evaluate and correlate the forest fire alerts and deforestation in the Amazon Forest in the state of Pará. The analyses were based on carrying out a diagnosis of forest fires and deforestation; the behavior of forest fires and deforestation over the last twenty years; the statistical relationship between deforestation and forest fires and their spatialization. This work identified that Pará is the state in the Legal Amazon with the highest occurrence of forest fires and deforestation. Deforestation in the four-year period Jan/2003-Dec/2006 showed a higher rate compared to the four-year periods Jan/2011-Dec/2018. A high correlation was found between forest fire alerts and increases in deforestation. There is a spatial relationship between cities with greater increases in deforestation and high numbers of fire alerts. In relation to the occurrence of forest fires and deforestation, the south of the state was the most critical region and the north had lower rates.

## Introduction

Globally, the reduction of deforestation combined with the role of forests as carbon sinks are key actions for mitigating climate change [1]. The largest tropical forest in the world is the Amazon, which plays a crucial role in maintaining the planet's climate. The Amazon Forest holds the greatest biodiversity on the planet and has a large part of its territory (60%) in Brazil [2]. The Brazilian Legal Amazon is an extensive territory of approximately 5,088,668.25 km$^2$ in which different factors influence the dynamics of forest fires in the region [3].

The primary consequences of forest fires are a decrease in biodiversity, soil degradation, climate change, and changes in the hydrological regime [4–7]. The destruction of tropical forests

**Funding:** This research was funded by the Coordenação de Aperfeiçoamento de Pessoal de Nível Superior (CAPES), Brazil, and the Conselho Nacional de Desenvolvimento Científico e Tecnológico (CNPq), Brazil.

**Competing interests:** The authors have declared that no competing interests exist.

by fires or deforestation is influenced by several environmental and anthropogenic factors distributed unevenly in a given area [8]. Forest fires result from complex interactions between human, geographic, and climatic conditions [9].

The Brazilian Amazon Forest has been threatened by increased livestock activities, forest fires, and illegal logging. Deforestation for the expansion of agriculture and livestock is one of the main causes of forest fires in South America, as it removes vegetation cover and increases the availability of combustible [10].

Pará is the second largest state in Brazil in terms of territorial extension and is home to a significant portion of the Amazon Forest, it is estimated that more than half of the state is covered by the Forest [2]. Unfortunately, this region also faces challenges related to illegal deforestation and forest fires [11].

Contrary to this need to prevent impacts on the Amazon Forest, financial resources for the protection and maintenance of this ecosystem have become increasingly scarce [12]. This fact has required public agents to take increasingly assertive and efficient measures to mitigate impacts on the forest.

Currently, the Brazilian government has two tools in the public domain and free of charge *PRODES* and *BDQueimadas*, to monitor and make decisions to combat deforestation and forest fires respectively, both are managed by the INPE (National Institute for Space Research) [2].

The *PRODES* project monitors the deforestation of forest formations in the Legal Amazon, using satellite images, techniques for estimating deforestation rates, and forest maps [13]. *BDQueimadas* is a database of fires and forest fires in Brazil, which is fed by information from different sources, including satellites and heat detection systems [14].

There are several works that use the *PRODES* and *BDQueimadas* databases to spatialize environmental impacts and guide public policies to combat deforestation and forest fires [10–15].

The *PRODES* and *BDQueimadas* databases have already been studied in the scientific literature and allow evaluation of the spatial overlap between fires and deforested areas, identifying patterns and interactions between these events, improving monitoring and control strategies, and providing a scientific basis for studies on forest dynamics in the Amazon [7, 8]. This comparison contributes to a deeper understanding of forest degradation processes and helps in the conservation and protection of this important biome. However, studies correlating both databases are scarce in the scientific literature.

The authors [16] highlight that studying the influence of forest fires and their spatiotemporal trends, associated with climatic factors, is extremely relevant for the conservation of biological and water resources in the Amazon Forest, especially in the state of Pará, where the highest values of deforestation of the Legal Amazon. Furthermore, fires can change the local microclimate, affecting the health of forestry workers in these regions [17, 18].

Therefore, this work aimed to evaluate the forest fire alerts and deforestation in the Amazon Forest in the state of Pará, Brazil, by making statistical inferences and correlating both databases.

## Materials and methods

### Characterization of the study area

The studied area corresponds to the legal Amazon limited to the state of Pará, Brazil. This region lies approximately between 0˚ 41' N and 9˚ 35' S latitude, and between 46˚ 43' W and 56˚ 8' W longitude. The authors [8] reinforce that the target area of study has the highest rates of deforestation and high numbers of forest fires.

The region in question has a rainy tropical climate of the Aw type, in the Köppen classification, with high temperatures and humidity and considerable rainfall fluctuations over time [19]. It has high precipitation throughout the year, varying between 2,000 and 3,500, and average annual temperatures ranging from 25°C to 28°C millimeters [20]. This fact contributes to the formation of the Amazon Forest present in the region, although there are variations due to the geographic diversity of the state, which includes forest areas, plains, mountains, and coastal regions.

The predominant vegetation in the region is the Submontane Dense Ombrophilous Forest, which occupies areas of mountainous relief and plateaus with medium-deep soils [21]. However, human activities have caused significant changes in forest areas, especially in the last three decades, with the frequent occurrence of forest fires, caused by human actions, such as burning to clear land. These fires have caused significant damage to natural vegetation, resulting in the loss of forest biodiversity [8].

## Forest fire alerts database

The data on forest fire alerts were taken from the *BDQueimadas* program, developed and maintained by the INPE (National Institute for Space Research) [2], which stores information on burning and forest fires recorded throughout the Brazilian territory. This database is public domain and free of charge.

Burnings were considered intentional practices, generally conducted by humans, with objectives such as clearing agricultural areas, renewing pastures, or managing ecosystems. Forest fires were considered uncontrolled events that occur in natural areas, with the potential to cause significant damage.

The *BDQueimadas* database is updated daily and records hot spots from 1998 to the present day. This database is a valuable tool for monitoring and analyzing the occurrence of burning and forest fires over time, and is fed by information collected from various remote sensors, such as satellites, radars, and other data sources. The information collected includes the date and time of registration, geographic location, and other relevant characteristics, constituting an important tool to be used in scientific research [12].

## Deforestation database

The deforestation data was taken from *PRODES* (Project for Monitoring Deforestation in the Legal Amazon by Satellite) and is also maintained by INPE. It contains information about deforestation in the Brazilian Legal Amazon, which covers nine states in the country's Amazon region. The database is in the public domain, free of charge and updated annually, provides information on deforestation areas detected by satellite [8].

Information in the *PRODES* database includes data on geographic location, extent of deforested area, type of previous forest cover, and deforestation rate. The data is collected by environmental monitoring satellites, which allows INPE to monitor deforestation continuously and update the database annually.

*PRODES* uses images from LANDSAT class satellites with 20 to 30 meters of spatial resolution and a 16-day revisit rate, in a combination that seeks to minimize the problem of cloud cover and guarantee interoperability criteria. Images from the American LANDSAT-5/TM satellite were, historically, the most used by the project, but images from the CCD sensor on board CBERS-2/2B, satellites from the Sino-Brazilian remote sensing program, were widely used. *PRODES* also made use of LISS-3 images from the Indian IRS-1 satellite and images from the English UK-DMC2 satellite. It currently makes massive use of images from LANDSAT 8/OLI, CBERS 4, and IRS-2. Regardless of the instrument used, the minimum area mapped by *PRODES* is 6.25 hectares [22].

### Spatial and statistical analysis of deforestation and forest fires data

Deforestation and Forest Fire data were compiled into tables, graphs, and maps with the help of statistical software: Graphpad Prism 8, Stastistic 7.0, Qgis 3.2, Origin Pro 8.0, and Office Package.

After extracting information from *PRODES* and *BDQueimadas*, the data was processed and interpreted according to the following methodological procedure:

• Diagnosis of forest fires and deforestation in the State of Pará

This analysis consisted of working with all the data, from *PRODES* and *BDQueimadas*, accumulated regarding the Legal Amazon. The increases in deforestation were added from the period from 1988 to 2022. The total forest fire alerts were quantified in the period from 1988 to 2022. The data was compiled in a table showing the extent of deforestation and forest fire alerts, both as a percentage in relation to Legal Amazon.

The present study was based on what was proposed by our colleagues [15], these types of data provide a comprehensive view of the extent and location of deforested areas affected by fires in the Legal Amazon. The analysis allows us to visualize the impact of the state area in Pará in comparison with other territories belonging to the Legal Amazon in Brazil.

• The behavior of forest fire alerts and deforestation in the State of Pará

The study was based on the model proposed by our colleagues [12] which states that when statistically comparing data from accumulated periods of time, environmental variables such as El Niño and La Niña are mitigated. Therefore, this study subdivided the Forest Fires and Deforestation data from the last twenty years (January 2003 –December 2022), into groups every four years.

Descriptive statistics were used to evaluate the behavior of variables through Mean, Standard Deviation, Coefficient of Variation, and Coefficient of Determination. Subsequently, ANOVA (Analysis of Variance) and the Student–Newman–Keuls test statistic were applied for multiple comparisons and their significance at 5%. The Graphpad Prism 8 and Stastistic 7.0 software were used to analyze and display the results.

• Relationship between deforestation and forest fires and its spatialization in the State of Pará

To investigate the correlation between the variable's deforestation and forest fires, the Origin Pro 8.0 software was used. A correlation statistical analysis was carried out between data on fire alerts and increases in deforestation, recorded in the municipalities of the state of Pará, in the years 2006, 2010, 2014, 2018, and 2022 at a significance level of 0.01. The data was compiled and graphs were created for each scenario with the Graphpad Prism 8 software. Tables expressing the similarity of locations and occurrences of forest fires and deforestation were constructed.

## Results

The deforestation data (*PRODES*) and forest fires (*BDQueimadas*) were worked on the basis of 144 municipalities present in the state of Pará [23] in different time scenarios, a fact that provided an assessment of the behavior of these variables.

### The forest fire scenario and deforestation in the state of Pará

All data on the increase in deforestation and forest fire alerts in the Legal Amazon were compiled and presented in Table 1. In the historical series of the two databases, *PRODES* and *BDQueimadas*, the state of Pará is the leader in Forest Fires and Deforestation. Therefore, the

**Table 1. Deforestation rate in the Brazilian Legal Amazon and percentage of forest fires.**

| State | Deforestation (Km²)* | Forest fire alerts** |
|---|---|---|
| Pará | 166,774.00 (34.61%) | 27% |
| Mato Grosso | 152,078.00 (31.56%) | 26% |
| Rondônia | 6,103.00 (13.72%) | 10% |
| Amazonas | 33,384.00 (6.93%) | 6% |
| Maranhão | 26,374.00 (5.47%) | 15% |
| Acre | 17,508.00 (3.63%) | 4% |
| Roraima | 9,188.00 (1.91%) | 1% |
| Tocantins | 8,790.00 (1.82%) | 9% |
| Amapá | 1,670.00 (0.35%) | 1% |
| **Legal Amazon** | **421,869.00 (100%)** | **100%** |

*Accumulated from the total *PRODES* database from 1988 to 2022.

**Accumulated from the total *BDQueimadas* database from 1998 to 2022.

prominent negative role and the need for a greater understanding of the dynamics of these variables are observed.

At this juncture, it's imperative to underscore that despite not being the largest geographically in the Legal Amazon, the state of Pará takes the lead in both forest fire outbreaks and deforestation

## The behavior of forest fire alerts and deforestation in the state of Pará

In relation to deforestation in the state of Pará, the four-year period Jan./2003-Dec./2006 presented a higher (p<0.05) rate compared to Jan./2011-Dec.-2014 and Jan./2015-Dec./2018, the other periods analyzed showed no difference (Fig 1A).

The Legal Amazon is an extensive territory made up of nine states in Brazil, the state of Pará has a high representation in deforestation in this region, as it reached a rate of 50.52% of all deforestation in the Legal Amazon in the four-year period Jan./2007-Dec./2010 (Fig 1B).

Analysis of the database of the last 20 years of forest fire alerts showed no difference (p<0.05) between four years (Fig 1C). However, when studying the behavior of regression curves for forest fires and deforestation, similar behavior is observed (Fig 1D).

The notable fluctuation in deforestation rates over time in the state of Pará emphasizes the necessity for a comprehensive spatial and statistical correlation analysis to discern the underlying factors at play.

The descriptive statistics of Forest Fires and Deforestation are presented in Table 2. The samples displayed heterogeneous behavior, characterized by a high standard deviation, reflecting a moderate coefficient of variation across the databases for the periods studied. The regression models employed to analyze forest fires and deforestation yielded a coefficient of determination exceeding 90%. Therefore, it is crucial to investigate whether areas with forest fire outbreaks are related to the deforestation recorded.

## Relationship between deforestation and forest fires and its spatialization in the state of Pará

It is possible to notice the correlation (p<0.01) between forest fire alerts and increases in deforestation for the years 2006, 2010, 2014, 2018, and 2022 (Fig 2A–2E). Therefore, there is a spatial relationship between cities in the state with greater increases in deforestation and the numbers of forest fire alerts.

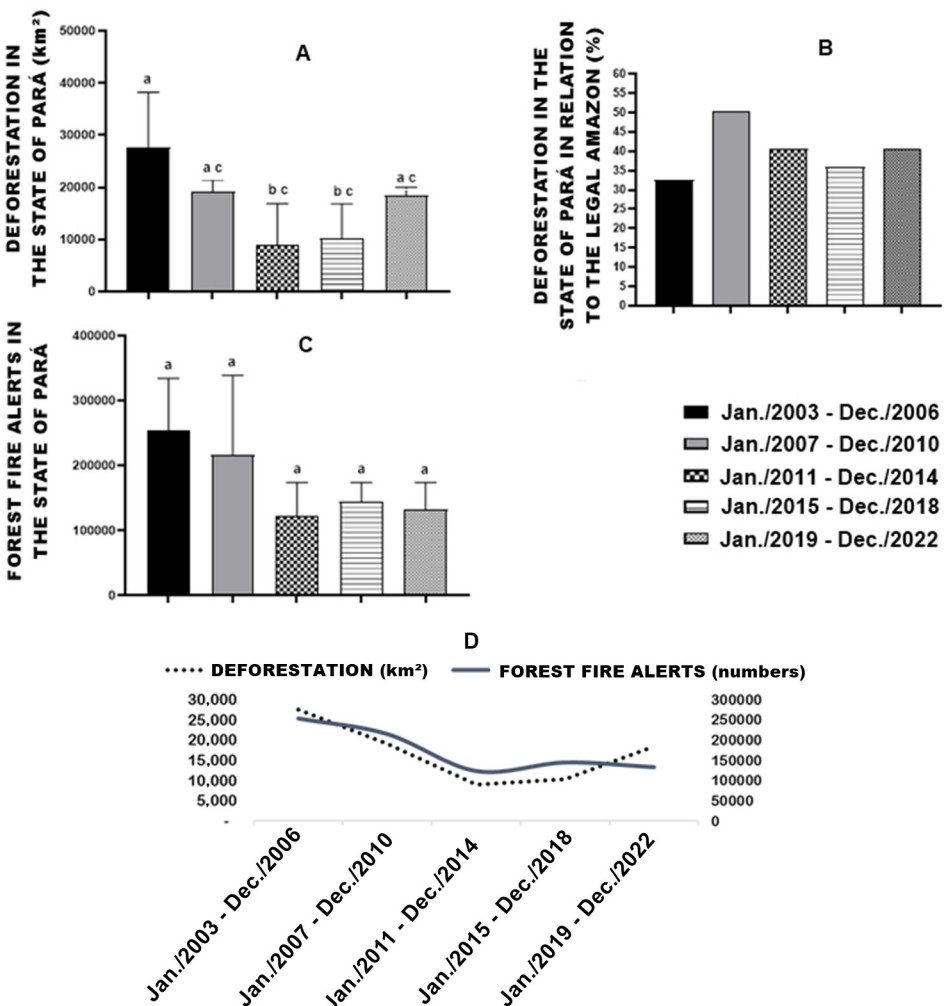

**Fig 1.** Deforestation in the state of Pará (A). Percentage of deforestation in the state of Pará in relation to the legal Amazon (B). Forest fire alerts in the state of Pará (C). Regression curves, deforestation, and forest fire alerts (D). Means followed by the same letter do not differ from each other using the Newman Keuls post hoc test at 5% probability. Statistically significant differences were observed in deforestation, whereas forest fire alerts did not show statistically significant differences. Regression equations for Forest Fire Incidents: $y = 12002x^2 - 103417x + 352539$, and for Deforestation: $y = 3184x^2 - 21815x + 47323$. The x-axis denotes time intervals from Jan/2003 to Dec/2022.

These locations were described and quantified in Tables 3 and 4. The south of the state of Pará has the highest rates of deforestation and a high presence of forest fire alerts, with emphasis on the municipalities of São Félix do Xingu and Altamira, which lead the ranking. The north of the state of Pará had the lowest rates of forest fires and deforestation.

Hence, the statistical correlation analysis depicted in Fig 2 elucidates the connection between forest fire alerts and deforestation. This interaction is visually represented in Tables 3 and 4.

**Table 2. Descriptive statistical analysis of forest fires and deforestation in the state of Pará.**

| Variable | Average | σ (Standard Deviation) | CV (%) | $R^2$ |
|---|---|---|---|---|
| Deforestation | 16,902.20 ($Km^2$) | 6,745.32 ($Km^2$) | 39.9% | 0.95 |
| Forest fire | 174,306 (Focos) | 5,1643.41 (Focos) | 29.63% | 0.90 |

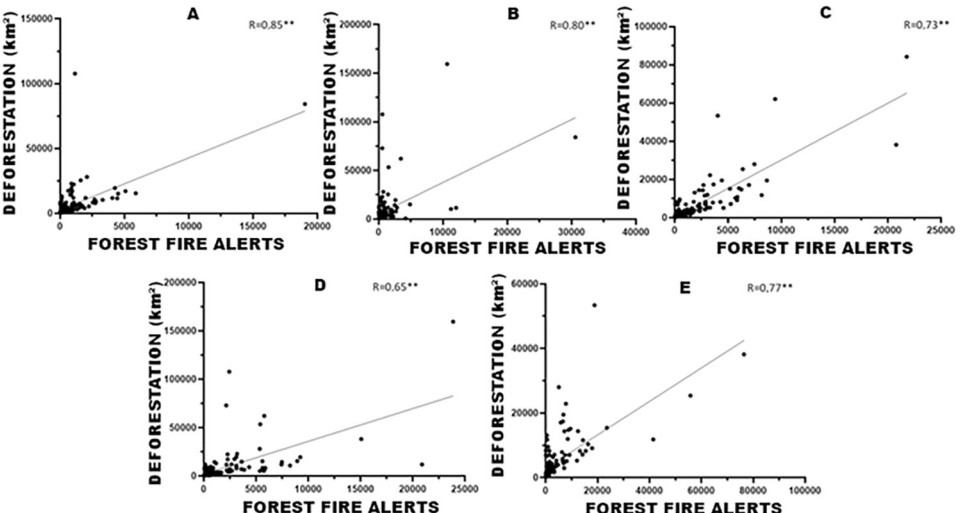

**Fig 2.** Correlations between forest fire outbreaks and increases in deforestation in the years 2006 (A), 2010 (B), 2014 (C), 2018 (D) and 2022 (E). **p<0.01. The statistical correlation coefficients highlight a pronounced positive association between forest fire outbreaks and the escalation of deforestation over the analyzed periods.

## Discussion

The main biome of the Amazon basin is the tropical moist evergreen forest known as the Tropical Amazon Forest. It is characterized by a predominantly closed environment with emergent trees and structured layers, intolerant to fire [24].

The Brazilian Legal Amazon is a region of extreme importance in terms of environmental preservation and biodiversity, made up of nine important states. Among these states, Amazonas stands out as the largest in territorial extension. However, it is important to highlight that Pará, although smaller in size, has a high representation in deforestation and forest fire alerts in the region. These numbers contributed to the fact that in the period from 1998 to 2018, Brazil stood out for having more than 60% of fire alerts in South America [10].

In this research, during the period from 1988 to 2022, it is estimated that an area of approximately 16,677,400 $km^2$ has been deforested in the state of Pará. These impressive numbers reveal the magnitude of deforestation occurring in Pará associated with the high occurrence of forest fires over the decades, a fact already diagnosed in other studies [8, 12, 15].

In 2020, deforestation reached the highest rate in the last 10 years, this increase is due to the use of fire to open pastures in the region. The slash-and-burn method alters the physical,

**Table 3. Cities that lost the most forest cover between 2006 and 2022.**

| Number | City | Total ($km^2$) |
|---|---|---|
| 1° | São Félix do Xingu | 7162.39 |
| 2° | Altamira | 6783.06 |
| 3° | Novo Repartimento | 3511.81 |
| 4° | Novo Progresso | 3008.73 |
| 5° | Pacajá | 2627.23 |
| 6° | Itaituba | 2649.33 |
| 7° | Portel | 2029.76 |
| 8° | Marabá | 1675.01 |
| 9° | Anapu | 1594.48 |
| 10° | Uruará | 1564.18 |

Table 4.  Cities that presented the most forest fire alerts between 2006 and 2022.

| Number | City | Forest fire alerts (%) |
|---|---|---|
| 1° | São Félix do Xingu | 10.49 |
| 2° | Altamira | 9.22 |
| 3° | Novo Progresso | 6.06 |
| 4° | Cumaru do Norte | 4.07 |
| 5° | Pacajá | 3.71 |
| 6° | Portel | 3.55 |
| 7° | Novo Repartimento | 2.94 |
| 8° | Itaituba | 2.87 |
| 9° | Santa Maria das Barreiras | 2.64 |
| 10° | Uruará | 2.46 |

chemical, and biological properties of the soil and favors the initial establishment of pastures in the Amazon region. Profound changes may be occurring due to the invasion of flammable grasses after forest fires, promoting subsequent fires and modifying the floristic dynamics of the region [25].

In this sense, the development of strategies to avoid environmental degradation promoted by forest fires and deforestation requires understanding the dynamics of the people who control their occurrence spatially and over time [26].

In the period from 2001 to 2020, the state of Pará recorded the highest rates of deforestation and high numbers of forest fires. This fact became evident in this study when analyzing the periods from January 2003 to December 2022. Forest fires are becoming a dominant natural disaster and a threat to forest ecosystems around the world, with the spread of a forest fire is the result of a complex interaction between climate, topography (elevation, slope, aspect), forest type, and human activities [9].

Actions to combat the occurrence of anthropogenic forest fires are extremely necessary to reduce the deforested area. Consequently, quantifying contributing geoenvironmental and human factors is essential to understand the underlying causes and predict future occurrences of wildfires, as the behavior of the regression curves in this study is close.

The linear correlation analysis of these distributions shows a close rain of points, a fact that exposes a link between the occurrence of forest fire alerts and an increase in increase in deforestation. Deforestation and fires have significant impacts on rainfall variability and land surface temperature in the Brazilian Amazon, they reduce forest evapotranspiration, which leads to a consequent decrease in precipitation, favoring subsequent forest fire events and increased increments of deforestation [8].

Therefore, it is essential that public policies to combat deforestation are aligned with the prevention of forest fires and the understanding of the different anthropogenic uses of fire in the region. Although climate plays an important role in the fires that hit the Amazon, human action also has a great influence on the occurrence of fires [27].

This research demonstrates that the regions with the highest forest fire alerts recorded in 2006, 2010, 2014, 2018, and 2022, presented higher rates of accumulated deforestation increases until 2022. When comparing the percentage of forest fire alerts with deforestation in the cities, the cities of São Félix do Xingu and Altamira are leaders in the ranking of forest fires and deforestation in the period analyzed, a behavior that is evident in other cities on the list.

This fact was also diagnosed by the work of the authors [8] in which he examined the period from 2001 to 2020 the "arc of deforestation", a region that extends close to the south of the state of Pará, stood out as a critical region due to the loss of forest cover and high incidence of

fires in the Amazon Brazilian. This region is where the cities of São Félix do Xingu and Altamira, diagnosed in the present study, are located.

Deforestation and fires are more frequent in areas of primary forest with road access. Since such areas have easy access and high economic value in the region. In this same work, there is a comparison of rural settlements with indigenous reserves, these had a total area of 40% affected by forest fires and 17% deforested, while indigenous reserves stood out for their important role in protecting forests [26].

It is evident that fighting forest fires entails high costs that can be measured mainly in terms of loss of human and animal lives, investments in firefighting resources, damage to the environment, and financing the restoration of affected areas [28]. Using satellite images combined with studies correlating deforested areas and forest fires can optimize public actions and define better strategies, reducing costs in the process of preservation and sustainable management of the Amazon forest in the state of Pará.

The deforestation and forest fire analyses presented in this study showcase an innovative, efficient, and swift methodology for guiding public interventions. Nonetheless, it is crucial that forthcoming discussions delve into studies such as those concerning carbon emissions [29] and phytoclimatic relationships in the Amazon region [30].

## Conclusions

This study indicates that there is a correlation between forest fire data and deforestation. It is not recommended to combat deforestation without efficient measures against forest fire alerts in the Amazon of the state of Pará.

The study corroborates a methodology for evaluating fires and deforestation from the *PRODES* and *BDQueimadas* databases. Factors that highlight the heterogeneity of the sampled data should be studied in future research, such as rainfall rates, different income classes, the outbreaks presence of indigenous reserves, and others.

Therefore, it is essential that efforts to combat forest fires and consequent deforestation must be concentrated in the southern region of the state of Pará, highlighting the cities of São Félix do Xingu and Altamira, in relation to the northern region of the state, which was less affected and maintained greater forest coverage.

It is paramount to emphasize the significance of conducting similar studies within the same geographical area, utilizing varied methodologies. This approach is essential for validating the quantitative findings and correlations between deforestation and forest fires as identified in this study. Moreover, such an approach substantially enhances the robustness and reliability of the gathered information, thereby furnishing a sturdy framework to inform environmental conservation policies and practices in the region.

## Supporting information

**S1 Data. Supplementary data for creating graphs on forest fire alerts and deforestation in the Legal Amazon, Brazil.**
(XLSX)

## Acknowledgments

Federal Institute of Education, Science and Technology of Pará (IFPA) and Federal University of Viçosa (UFV).

## Author Contributions

**Conceptualization:** Cássio Furtado Lima, Fillipe Tamiozzo Pereira Torres, Luciano José Minette, Arthur Araújo Silva, Francisco de Assis Costa Ferreira.

**Data curation:** Cássio Furtado Lima, Fernanda Araujo Lima, Michel Keisuke Sato, Francisco de Assis Costa Ferreira.

**Formal analysis:** Cássio Furtado Lima, Fernanda Araujo Lima.

**Funding acquisition:** Francisco de Assis Costa Ferreira.

**Investigation:** Cássio Furtado Lima, Luciano José Minette, Fernanda Araujo Lima, Francisco de Assis Costa Ferreira.

**Methodology:** Cássio Furtado Lima, Fernanda Araujo Lima, Francisco de Assis Costa Ferreira.

**Project administration:** Cássio Furtado Lima, Bruno Leão Said Schettini.

**Resources:** Mateus Xavier Lima Machado.

**Software:** Cássio Furtado Lima, Michel Keisuke Sato, Mateus Xavier Lima Machado.

**Supervision:** Fillipe Tamiozzo Pereira Torres, Luciano José Minette, Arthur Araújo Silva, Bruno Leão Said Schettini.

**Validation:** Cássio Furtado Lima, Fillipe Tamiozzo Pereira Torres, Luciano José Minette, Roldão Carlos Andrade Lima, Michel Keisuke Sato, Arthur Araújo Silva.

**Visualization:** Mateus Xavier Lima Machado.

**Writing – original draft:** Cássio Furtado Lima, Roldão Carlos Andrade Lima, Michel Keisuke Sato, Arthur Araújo Silva.

**Writing – review & editing:** Roldão Carlos Andrade Lima, Mateus Xavier Lima Machado.

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
