## [Decision Letter · Decision Letter 0]

15 Feb 2024

PONE-D-23-44049Is there a relationship between forest fires and deforestation in the Brazilian Amazon?PLOS ONE

Dear Dr. Andrade Lima,

Thank you for submitting your manuscript to PLOS ONE. After careful consideration, we feel that it has merit but does not fully meet PLOS ONE’s publication criteria as it currently stands. Therefore, we invite you to submit a revised version of the manuscript that addresses the points raised during the review process.

One of the reviewers find the paper good enough, while we were waiting for other revisions that never materialize. Then I inspected the paper by myself and agree with the one reviewer's points. Specifically, please you need to clarify better the findings and statistics considered. Also, please give clear responses to the general  points raised by the reviewer.

We look forward to receiving your revised manuscript.

Kind regards,

Daniel Capella Zanotta

Academic Editor

PLOS ONE

2. We note that Figures 1 and 4 in your submission contain [map/satellite] images which may be copyrighted. All PLOS content is published under the Creative Commons Attribution License (CC BY 4.0), which means that the manuscript, images, and Supporting Information files will be freely available online, and any third party is permitted to access, download, copy, distribute, and use these materials in any way, even commercially, with proper attribution. For these reasons, we cannot publish previously copyrighted maps or satellite images created using proprietary data, such as Google software (Google Maps, Street View, and Earth). For more information, see our copyright guidelines: http://journals.plos.org/plosone/s/licenses-and-copyright.

1. You may seek permission from the original copyright holder of Figures 1 and 4 to publish the content specifically under the CC BY 4.0 license. 

Reviewers' comments:

Reviewer's Responses to Questions

**Comments to the Author**

1. Is the manuscript technically sound, and do the data support the conclusions?

Reviewer #1: Partly

2. Has the statistical analysis been performed appropriately and rigorously? 

Reviewer #1: N/A

3. Have the authors made all data underlying the findings in their manuscript fully available?

Reviewer #1: No

4. Is the manuscript presented in an intelligible fashion and written in standard English?

Reviewer #1: Yes

5. Review Comments to the Author

Reviewer #1: It's understandable to question the need for repeating studies, especially when there are existing investigations on a similar topic. Conducting research on the same or a related subject can serve various purposes, including:

Verification and Validation: Replicating studies allows researchers to validate and verify the results of previous research, contributing to the robustness of scientific findings.

Contextual Adaptation: Environmental and geographical contexts vary, and what holds true in one region may not necessarily apply directly to another. Replicating studies in different locations can help understand how findings might be influenced by specific conditions.

Temporal Changes: Over time, environmental conditions, land use patterns, and other factors may change. Repeating studies at different points in time helps capture these changes and assess how they impact the relationship between variables.

Methodological Refinement: Researchers may use similar methodologies but with improvements or modifications to address limitations identified in previous studies, enhancing the precision and reliability of the results.

Novel Contributions: While a study may have a similar overarching theme, the specific research questions, methodologies, and objectives might differ. New studies can bring novel perspectives, insights, or data that contribute to the cumulative knowledge in the field.

Regarding the specific studies you mentioned (DOI: 10.1007/s00267-015-0447-7 and DOI: 10.1016/j.apgeog.2011.10.013), it's essential to consider the context, objectives, and focus of your research. If your study aims to address specific aspects or nuances not covered by existing research, it can still make a valuable contribution to the scientific literature. Additionally, discussions with colleagues or experts in the field can help refine your research questions and approach, ensuring that your work adds meaningful insights to the existing body of knowledge.

6. PLOS authors have the option to publish the peer review history of their article (what does this mean?). If published, this will include your full peer review and any attached files.

Reviewer #1: No

---

## [Author Response · Author response to Decision Letter 0]

23 Mar 2024

Author's Response to Reviewers Comments 3

Dear Editor and Reviewer.

We appreciate the speed in the manuscript evaluation process and the positive response to the scientific merit of the work. I hope that after these new considerations, the work fits within the highlighted criteria.

Point 1: We note that you state the following regarding your map data: "We appreciate the registrations made and the concern with intellectual property regarding Figures 1 and 4. It is important to highlight our dedication to following all legal, ethical, and compliance guidelines expressed in licenses, such as CC BY 4.0. We want to make it clear that these figures are the intellectual property of the authors, that is, we created them using the QGIS 3.2 software. Therefore, we did not remove the figures from the MAPBIOMAS website.

Spatialization and vegetation cover data are made available free of charge to the public by MAPBIOMAS, promoting transparency and allowing free access to crucial information for public policies, environmental monitoring, and territorial planning. What we did was interpolate this free data using the Kernel density estimation method, available in the QGIS 3.2 data processing tool (described in the methodology).

Therefore, the aforementioned institution will not sign this agreement, as it is not responsible for creating the images in this work, that is, the aforementioned images were not taken from MAPBIOMAS, being the exclusive intellectual property of the authors of this work."

However, the data from MAPBIOMAS is still copyrighted under CC BY-SA and as you used the data to create your maps, permission is still needed from the copyright holder to publish the map images using the copyrighted data.

Response 1: We are grateful for the records made and the care shown with the intellectual property of Figures 1 and 4. We would like to highlight that these figures are the intellectual property of the authors and that the uses of the aforementioned public data, as well as documents already forwarded, are by common consent in the national territory of Brazil. 

However, to meet the requirements of this prestigious journal, we decided to remove the respective figures from the manuscript.

It is important to emphasize that this modification will not compromise the quality or the identification of the correlation between forest fires and deforestation, since the statistical and spatial conclusions remain unchanged.

Kind regards.

---

## [Editor Report · Decision Letter 1]

15 Jun 2024

Is there a relationship between forest fires and deforestation in the Brazilian Amazon?

PONE-D-23-44049R1

Dear Dr. Andrade Lima,

We’re pleased to inform you that your manuscript has been judged scientifically suitable for publication and will be formally accepted for publication once it meets all outstanding technical requirements.

Kind regards,

Daniel Capella Zanotta

Academic Editor

PLOS ONE

Additional Editor Comments (optional):

After a few tries, I couldn't have feedback from the previous reviewer about the revision made by the authors, therefore I did the validation of the revision by my own.

I comply with the replies provided by the authors since it is soundness. Few changes were made in the manuscript, but the reviewer indeed did not require many substantial ones.

Then, I proceed with the acceptance.

Reviewers' comments:

Noone

---

## [Editor Report · Acceptance letter]

20 Jun 2024

PONE-D-23-44049R1 

PLOS ONE

Dear Dr. Andrade Lima, 

I'm pleased to inform you that your manuscript has been deemed suitable for publication in PLOS ONE. Congratulations! Your manuscript is now being handed over to our production team.

Kind regards, 

on behalf of

Dr. Daniel Capella Zanotta 

Academic Editor

PLOS ONE